# A Minimal Physiologically Based Pharmacokinetic Model to Characterize CNS Distribution of Metronidazole in Neuro Care ICU Patients

**DOI:** 10.3390/antibiotics11101293

**Published:** 2022-09-22

**Authors:** Alexia Chauzy, Salim Bouchène, Vincent Aranzana-Climent, Jonathan Clarhaut, Christophe Adier, Nicolas Grégoire, William Couet, Claire Dahyot-Fizelier, Sandrine Marchand

**Affiliations:** 1UFR de Médecine-Pharmacie, Université de Poitiers, Inserm U1070, 86073 Poitiers, France; 2Clinical Pharmacology Department, Menarini Stemline, 50131 Florence, Italy; 3Laboratoire de Pharmacocinétique-Toxicologie, CHU de Poitiers, 86021 Poitiers, France; 4Service d’Anesthésie-Réanimation et Médecine Périopératoire, CHU de Poitiers, 86021 Poitiers, France

**Keywords:** physiologically based pharmacokinetic (PBPK), central nervous system, blood–brain barrier (BBB), blood-cerebrospinal fluid barrier (BCSFB), modelling and simulation, neuro care ICU patients

## Abstract

Understanding antibiotic concentration-time profiles in the central nervous system (CNS) is crucial to treat severe life-threatening CNS infections, such as nosocomial ventriculitis or meningitis. Yet CNS distribution is likely to be altered in patients with brain damage and infection/inflammation. Our objective was to develop a physiologically based pharmacokinetic (PBPK) model to predict brain concentration-time profiles of antibiotics and to simulate the impact of pathophysiological changes on CNS profiles. A minimal PBPK model consisting of three physiological brain compartments was developed from metronidazole concentrations previously measured in plasma, brain extracellular fluid (ECF) and cerebrospinal fluid (CSF) of eight brain-injured patients. Volumes and blood flows were fixed to their physiological value obtained from the literature. Diffusion clearances characterizing transport across the blood–brain barrier and blood–CSF barrier were estimated from system- and drug-specific parameters and were confirmed from a Caco-2 model. The model described well unbound metronidazole pharmacokinetic profiles in plasma, ECF and CSF. Simulations showed that with metronidazole, an antibiotic with extensive CNS distribution simply governed by passive diffusion, pathophysiological alterations of membrane permeability, brain ECF volume or cerebral blood flow would have no effect on ECF or CSF pharmacokinetic profiles. This work will serve as a starting point for the development of a new PBPK model to describe the CNS distribution of antibiotics with more limited permeability for which pathophysiological conditions are expected to have a greater effect.

## 1. Introduction

The central nervous system (CNS) is protected by physiological barriers controlling xenobiotic penetration. To act at the CNS level, drugs must have properties allowing them to cross the blood–brain barrier (BBB). Thus, compounds with limited CNS distribution are discarded during the development process. In vitro models of the BBB [1] have been designed to address this issue at the early development stage, as well as physiologically based pharmacokinetic (PBPK) models that allow the characterization in vivo of CNS distribution and interspecies extrapolations [2,3,4,5]. However, antibiotics are essentially used to treat infections localized in various organs or tissues, such as lung or peritoneal fluids, and CNS penetration is not considered a major decision criterion during antibiotic development. In fact, extensive diffusion in the CNS could rather be a problem for antibiotics such as fluoroquinolones, penicillins or carbapenems, which may induce undesirable CNS side effects [6,7,8]. Yet severe life-threatening CNS infections, such as nosocomial ventriculitis or meningitis, require rapid and aggressive antibiotic treatment that represents a real challenge [9]. These infections are increasingly caused by multidrug-resistant (MDR) bugs, and only a few antibiotics possess the appropriate microbiological spectrum to induce a rapid bactericidal effect [10,11,12]. Therapeutic options are then limited and may involve antibiotics with a priori limited CNS distribution, either because of their high hydrophilicity or/and affinity for active efflux pumps [13]. However, CNS distribution is likely to be modified in patients with brain damage and infection/inflammation due to alterations of physiological blood flows and volumes, as well as disruption of tight junctions and modifications of active transporter activity/expression. In patients with traumatic brain injury (TBI) or subarachnoid hemorrhage (SAH), an increase in intracranial pressure (ICP) due to a reduced cerebrospinal fluid (CSF) reabsorption [14] or/and a mechanisms leading to increased cerebral volumes, such as oedema or hematoma [15,16], is often observed. Moreover, in TBI and SAH patients, a decrease in cerebral blood flow [15,17,18] and an increase in permeability due to the breakdown of the BBB and the blood-cerebrospinal fluid barrier (BCSFB) have been reported [14,15]. Furthermore, patients are frequently equipped with external ventricular drains (EVD) that may also alter antibiotic exposure in CNS. Therefore, CNS distribution of antibiotics in brain-damaged patients is driven by multiple important factors that are difficult to predict. Yet CSF concentrations obtained from the EVD collection bag, close to concentrations at the infection site, are frequently used to adjust the dosing regimen for a particular patient. Traditional compartmental modeling has been frequently conducted to characterize the CNS distribution of antibiotics in brain-damaged patients [19,20,21,22,23,24]. However, PBPK modeling might be more appropriate to identify the parameters influencing antibiotic CNS distribution and to predict the impact of their pathophysiological alterations. However, any given pathophysiological alteration will not have the same impact on the CNS distribution of all antibiotics, depending on their physico-chemical characteristics. For example, increased membrane permeability due to the disruption of tight junctions is expected to have a greater effect on low- than high-permeability compounds. The same type of comment holds true for active efflux pumps. Yet among the various published brain PBPK models, some consider exclusively the BBB and not the BCSFB [2,25], which constitutes a reasonable simplification considering the authors’ objective. However, in order to attain the benefit of determining CSF concentrations in brain-damaged patients, it is necessary to develop a model including a BCSFB in addition to the BBB since exchanges exist between CSF and brain extracellular fluid (ECF). Three such PBPK models have been described in the literature [26,27,28] and were used to create the one proposed in this article, considering that our main objective was to set up a minimal PBPK including the main parameters governing antibiotics’ CNS distribution, which is likely to be altered in brain damaged patients. As a first step, we have chosen to develop this model using metronidazole, an antibiotic with extensive CNS distribution simply governed by passive diffusion, for which we had relatively rich clinical data available, including CSF concentrations in patients equipped with EVD [29] and brain ECF concentrations in patients monitored with brain microdialysis [30].

## 2. Results

### 2.1. PBPK Analysis

The minimal PBPK model described unbound metronidazole PK profiles in plasma, brain dialysate after correction by in vivo probe recovery and an EVD collection bag well as can be observed from the individual plots (Appendix A) and the visual predictive checks (VPCs) (Figure 1). The parameter estimates were obtained with good precision and are summarized in Table 1. The diffusion of metronidazole across the BBB and the BCSFB, calculated from physiological and physicochemical data, was predicted to be rapid and passive (6.4 L/h and 3.2 L/h, respectively). The model fit was not improved when non-equal rate constants were used across both the BBB and BCSFB. Mean fold errors between predicted and observed AUC_Δt_ were close to 1 for the three compartments (Appendix A). Figure 2 shows the predicted concentration profiles in plasma and cerebral compartments obtained in a typical patient. Profiles in ECF and CSF predicted by the model were similar.

### 2.2. Sensitivity Analysis

#### 2.2.1. Impact of Brain Pathophysiological Changes

The metronidazole concentration profiles obtained after varying PS_ECF/CSF_, V_ECF_ and Q_brain_ were almost identical, as shown in Figure 3.

#### 2.2.2. Impact of EVD

The impact on model predictions after changing the values of Q_EVD_ within a range of 0.001–0.04 L/h is shown in Figure 4. PK profiles of metronidazole in CSF were not sensitive to the changes in Q_EVD_.

### 2.3. Evaluation of Drug Permeability across the BBB and BCSFB

The PS_ECF_ value calculated from system and drug-specific parameters using Simcyp^®^ (PS_calculated_) was quite similar to the in vitro-scaled PS_ECF_ value (PS_in vitro_) but somewhat higher than the value estimated by the model (7.1-fold) (Table 2). Similarly, PS_CSF,calculated_ was 8-fold higher than PS_CSF,estimated_. A ratio close to 2 (2.3) between PS_ECF_ and PS_CSF_ was predicted by the model. All other parameters of the model were almost identical (Appendix A). This resulted in similar predicted ECF and CSF concentration profiles regardless of the values used for the passive diffusion clearances (Appendix A).

## 3. Discussion

Metronidazole was selected as an appropriate compound to develop a minimal PBPK model, not only because data were available both in human brain ECF and CSF, which, as far as we know, is unique, but also because we previously showed that its distribution within brain ECF [30] and CSF [29] is extensive, in agreement with its lack of affinity for efflux transport systems at BBB and BCSFB, making this initial PBPK model relatively simple. In fact, metronidazole was shown to be an inhibitor but not a substrate of P-gp [31].

Metronidazole permeability across the BBB (PS_ECF_) was predicted from drug-specific parameters (MW and log P) and the BBB surface area. By comparison, the value given by Simcyp and used in the PBPK model (6.4 L/h) was similar to the value calculated using previously reported equations from the literature (6.3 L/h) [32] but slightly higher than the value obtained from PK-Sim, another PBPK platform (2.4 L/h). To validate the BBB permeability of metronidazole, the bidirectional passive permeability value used in the PBPK model was compared to the value determined from an in vitro Caco-2 model. The in vitro-scaled PS_ECF_ value (8.0 L/h) was quite close to the Simcyp prediction, indicating a good correlation between these two approaches. Moreover, the determination of an in vitro efflux ratio lower than 2 (Table 1A) confirmed that metronidazole was not a substrate of efflux transporters and that its brain distribution was exclusively passive. In the present study, the Caco-2 model was used as a surrogate BBB penetration model (Appendix B). Although this is a simple epithelial cell model routinely used in drug development for the prediction of intestinal absorption, it is also commonly used to evaluate the BBB permeability of drugs [33,34]. The use of a more laborious and expensive brain endothelial cell-based model that more closely reproduces the main BBB features and the expression of specific transporters did not seem essential in the case of metronidazole [35]. On the other hand, PS_CSF_ was assumed to be half of PS_ECF_, consistent with previous PBPK models [4,27]. To validate this hypothesis, PS_ECF_ and PS_CSF_ were estimated by the model, and although much lower values were obtained (Table 2), a ratio close to 2 was still observed, and similar CNS PK profiles were obtained (Appendix A).

Patients in both groups were suffering from neuro-trauma and CNS drug distribution characteristics observed in these patients could be altered. Thus, the impact of pathophysiological changes on PK profiles in brain ECF and CSF was explored by replacing PS_ECF/CSF_, V_ECF_ and Q_brain_ with values that can be found in patients suffering neurovascular diseases such as TBI and SAH. First, an a priori 2- to 5-fold increase in PS_ECF/CSF_ corresponding to the increase in the ratio between CSF and plasma albumin, reflecting a pathological increase in permeability of endothelial cells, was evaluated [36]. Since PS_ECF/CSF_ values were initially high, the increased permeability under pathological conditions had no impact on the metronidazole profiles in ECF and CSF (Figure 3a). 

Previous studies showed that the swelling process accompanying acute brain injury was mainly due to an increase in brain tissue water responsible for an increase in V_ECF_, whereas a reduction in V_CSF_ could be observed to compensate for the increase in brain-tissue water [37,38]. Therefore, an increase in V_ECF_ comparable to the percentage of swelling previously observed in head-injured patients with edema was tested (average 9% increase in brain volume up to 35%) [37]. V_ECF_ was allowed to vary within a narrow range of values from 5%, corresponding approximately to the threshold value (4.3%) associated with an increase in intracranial pressure [39], to 40%. Although this small increase in V_ECF_ may have important consequences from a pathophysiological point of view, it had no impact on metronidazole concentrations in ECF (Figure 3b).

Finally, the effect of several thresholds values of Q_brain_ corresponding to mild hypoperfusion or oligemia (70% of normal), penumbra (40% of normal) and critical ischemia (20% of normal) [40] on metronidazole brain PK were investigated. The simulations showed no impact of Q_brain_ on the metronidazole profiles in ECF and CSF, which is consistent with the fact that the brain distribution is permeability-limited due to the BBB and BCSFB with their tight junctions, and not perfusion-limited (Figure 3c). In fact, despite the decrease in Q_brain_, its value was still higher than that of PS_ECF/CSF_. 

Since TBI and SAH conditions did not significantly influence metronidazole PK profiles, it was not necessary to adjust the values of blood flows, cerebral volumes or diffusion parameters to capture the CNS distribution of metronidazole in the patients of the present study. However, for less lipophilic antibiotics or with higher molecular weight, and thus with low permeability, brain pathophysiological changes, in particular barrier disruption, are expected to have more impact on their CNS PK. Indeed, if we consider a hypothetical drug that is not a substrate for efflux transporters but has the same physicochemical characteristics as a low-permeability antibiotic such as cefotaxime (MW = 455 g/mol, log P = −1.5), higher and earlier peak concentrations should be observed in ECF and CSF when PS_ECF/CSF_ increased, although a linear increase is not expected (steady state concentrations increased from 9.8 to 12 mg/L in brain ECF and from 8.5 to 11.5 mg/L in CSF when PS_ECF/CSF_ is multiplied by 5) (Appendix A). However, the increase in V_ECF_ should result in a minor increase in ECF concentrations, and the decrease in Q_brain_ should have no impact on CNS PK profiles (Appendix A).

Yet, patients were not similar in both groups. From a pathophysiological point of view, the patients of the study were relatively comparable, but one may notice that CSF concentrations were measured in patients equipped with an EVD to compensate for brain hyper-pressure due to the obstruction of the CSF outflow (Q_sink_), which was not the case for patients equipped with microdialysis. Yet the fraction of metrodinazole dose recovered in the collection bag ranged from 0.1 to 0.4%, which has, therefore, a negligible impact on total clearance and brain PK profiles (Figure 4), especially since the sum of Q_sink_ and Q_EVD_ was assumed to be equal to the physiological value of Q_sink_. However, an effect of the EVD on CSF concentrations could be observed for drugs with lower passive permeability when Q_EVD_ is higher than the physiological value of Q_sink_ (Q_EVD_ > 0.024 L/h). In this situation, it was assumed that CSF reabsorption was totally blocked, and thus, Q_sink_ was set at 0 L/h, resulting in a potentially greater impact of the loss of the drug via the EVD on CSF PK profiles for drugs with low permeability (Appendix A).

A limitation of this study is the limited number of patients in each group. However, a multicenter clinical trial coordinated by our group is running to investigate CSF distribution of 9 distinct antibiotics in patients that are infected or not [41]. Some of these antibiotics are known to have a limited distribution within CSF due to high hydrophilicity responsible for low membrane permeability or/and affinity for efflux transport systems such as OAT-3, PEPT-2 or MRP4 for cephalosporin or carbapenem [13,42], which characteristic will be implemented in the present model. Furthermore, for each antibiotic, 25 patients will be recruited, and parameters known to have a potential effect on CSF distribution, mostly markers of inflammation (albumin, interleukin, cytokines), will be measured and tested as potential covariates explaining between-patients variability. Lastly, these CSF PK results will be used for PKPD modeling and dosing regimen optimization. 

## 4. Materials and Methods

### 4.1. Patients, Metronidazole Administration, Sample Collection and Quantification of Metronidazole Concentrations

This study was performed in the neurointensive care unit at the University Hospital of Poitiers (France) and was approved by the local ethics committee (CPP OUEST III, protocol 2008-003311-12). The study design has been previously described in detail [29,30]. Briefly, eight brain-injured patients were divided into two groups based on whether they were equipped with a microdialysis probe into one of the two frontal lobes of the brain tissue (*n* = 4) or an EVD into their lateral brain ventricles (*n* = 4). The demographic characteristics are detailed in Table 3. At admission, intracranial pressure was monitored in all patients, and CT scans were monitored during their hospitalization. Patients were included in the PK study after stabilization of intracranial pressure and neurological status. All patients received 500 mg of metronidazole (B Braun, Boulogne-Billancourt, France) every 8 h through a 30-min intravenous infusion to treat a lung infection.

The metronidazole pharmacokinetic study was conducted at a steady state after at least 2 days of treatment. Brain dialysates and CSF samples were collected, respectively, during microdialysis monitoring or via the EVD over 8 h at 0.5 h intervals during the first 2–4 h and at 1 h intervals for the remainder of the experiment. Serial blood samples (between 7 and 13 per patient) were collected simultaneously over the dosing interval in all patients. Plasma unbound concentrations were determined by ultrafiltration, and all samples were assayed by high-performance liquid performance (HPLC) with UV detection (310 nm) as previously described [30].

### 4.2. Population PK Analysis

The concentrations in plasma and CNS from all patients were modelled simultaneously using the non-linear mixed effect modelling approach in NONMEM 7.4 (ICON Development Solutions, Ellicott City, MD, USA). All model estimations were conducted with the first-order conditional estimation method with interaction (FOCE INTER). Model evaluation was based on the visual inspection of diagnostic plots (goodness of fit (GOF) and individual plots) [43], comparison of the objective function value (OFV) and relative standard errors (RSEs) of the parameter estimates. RSEs of parameters were obtained using the sampling importance resampling (SIR) procedure implemented in PsN [44]. Model predictive performance was assessed using VPCs (samples *n* = 1000) and by calculating fold errors between predicted and observed areas under the plasma, brain dialysate and CSF unbound concentration-time curves between two consecutive dosing at steady-state (AUC_Δt_). Predicted AUC_Δt_ were determined based on predicted individual PK profiles for each tissue, while observed AUC_Δt_ were calculated using the linear trapezoidal rule (Phoenix WinNonlin version 6.2; Certara USA, Inc., Princeton, NJ, USA).

A minimal PBPK modelling approach was used to describe the concentration-time profiles of metronidazole in plasma and CNS. In each patient, two individual unbound fraction values (fu) of metronidazole were calculated as the ratio of metronidazole concentrations in ultrafiltrates to corresponding total plasma concentrations. The mean value was used to convert total concentrations into unbound concentrations. The blood to plasma (B/P) ratio of metronidazole was estimated using the software tools PK-Sim (version 10.0, Open Systems Pharmacology, Bayer Technology Services, Leverkusen, Germany) based on fu, log P and haematocrit, as defined by Rodgers and Rowland [45] (Table 4) and used to convert measured unbound plasma concentrations into unbound blood concentrations in the model. Metronidazole elimination was implemented as a total blood clearance representing both its renal excretion and hepatic metabolism. The CNS was represented by three physiologically based compartments: the brain vasculature, the brain ECF and the brain CSF (Figure 5). The rest of the tissues were lumped as a single, perfusion-limited and well-stirred tissue compartment. The tissue compartments were connected via blood flow (Q) to the blood compartment in a closed-loop format. Volumes (V) and blood flows were fixed to their physiological value obtained from the literature (Table 4).

Metronidazole concentrations in the blood compartment (C_B_) were modelled as: (1)$$V_{B}\frac{{\mathrm{dC}}_{B}}{\mathrm{dt}}=\mathrm{Input}+f_{d}\mathrm{CO}\frac{C_{\mathrm{tissue}}}{K_{p}}+Q_{\mathrm{brain}}C_{\mathrm{brain},\mathrm{vasc}}-C_{B}\left(f_{d}\mathrm{CO}+Q_{\mathrm{brain}}+\mathrm{CL}\right)$$
where Input is the infusion rate (1000 mg/h) for the dutaion of the infusion (0.5 h), V_B_ is blood volume, CO is cardiac output blood flow, C_tissue_ is the metronidazole concentrations in (non-CNS) tissue compartment, f_d_ is the fraction of CO going to the tissue compartment, and K_p_ is the unbound tissue to blood partition coefficient. C_brain,vasc_ is the metronidazole concentrations in brain vasculature, Q_brain_ is the brain blood flow, and CL is the total blood clearance.

Metronidazole concentrations in non-CNS tissue compartment (C_tissue_) were modelled as:(2)$$V_{\mathrm{tissue}}\frac{{\mathrm{dC}}_{\mathrm{tissue}}}{\mathrm{dt}}=f_{d}\mathrm{CO}\left(C_{B}-\frac{C_{\mathrm{tissue}}}{K_{p}}\right)$$
where V_tissue_, the volume of the tissue compartment, was determined by V_B_ + V_tissue_ + V_brain,vasc_ + V_ECF_ + V_CSF_ = TBW due to physiological constraints, with V_brain,vasc_ representing the volume of the brain vasculature compartment, V_ECF_ the ECF brain tissue volume, V_CSF_ representing the cranial CSF volume and TBW representing the total body weight (Table 4) [46]. Similarly, due to physiological restrictions, total blood flow was ≤CO, implying that the sum of CO fractions going to the brain (f_d,brain_) and non-CNS tissue compartment (fd) was ≤1 (f_d_ + f_d,brain_ ≤ 1), with f_d,brain_ fixed to the physiological value calculated from $\frac{Q_{brain}}{CO}$.

Metronidazole concentrations in the brain vasculature compartment (C_brain,vasc_) were modelled as:(3)$$V_{\mathrm{brain},\mathrm{vasc}}\frac{{\mathrm{dC}}_{\mathrm{brain}\;\mathrm{vasc}}}{\mathrm{dt}}=Q_{\mathrm{brain}}C_{B}+{\mathrm{PS}}_{\mathrm{ECF}}C_{\mathrm{ECF}}+C_{\mathrm{CSF}}\left({\mathrm{PS}}_{\mathrm{CSF}}+Q_{\sin k}\right)-C_{\mathrm{brain},\mathrm{vasc}}\left(Q_{\mathrm{brain}}+{\mathrm{PS}}_{\mathrm{ECF}}+{\mathrm{PS}}_{\mathrm{CSF}}\right)$$
where C_ECF_ and C_CSF_ are ECF and CSF concentrations, respectively, PS_ECF_ and PS_CSF_ are passive permeability-surface area products for the BBB and BCSFB, respectively, and Q_sink_ is the CSF absorption rate (sink flow). Metronidazole is not thought to be actively transported; therefore, only passive diffusion across the brain barriers was considered. Passive diffusion clearance at the BBB (PS_ECF_) was determined from a combination of system- (surface area of BBB, SA_BBB_) and drug-specific parameters (log P and molecular weight), using the prediction option incorporated in Simcyp simulator (version 18.0, Certara, Sheffield, UK). PS_ECF_ was fixed to 6.4 L/h and PS_CSF_ was assumed to be half of PS_ECF_ (considering SA_BCSFB_ = 50% SA_BBB_) [27].

Metronidazole concentrations in the ECF compartment (C_ECF_) were modelled as:(4)$$V_{\mathrm{ECF}}\frac{{\mathrm{dC}}_{\mathrm{ECF}}}{\mathrm{dt}}=C_{\mathrm{brain},\mathrm{vasc}}{\mathrm{PS}}_{\mathrm{ECF}}-C_{\mathrm{ECF}}\left({\mathrm{PS}}_{\mathrm{ECF}}+Q_{\mathrm{bulk}}\right)$$
where Q_bulk_ is the bulk flow from brain ECF to CSF. The bulk flow of ECF is a physiological convection process and was assumed to be the major mechanism of drug transfer between ECF and CSF.

Brain dialysate concentrations (C_ECF,dialysate_), collected as fractions during various time intervals, were described by the integral over each collection interval, represented by t_1_ and t_2_ [47]:(5)$$C_{\mathrm{ECF},\mathrm{dialysate}}=\overset{t_{2}}{\underset{t_{1}}{\int }}C_{\mathrm{ECF}}\mathrm{dt}$$

In the model, C_ECF,dialysate_ corresponds to the dialysate concentrations corrected by the in vivo probe recovery determined for each patient using a retrodialysis-by-drug method, as previously described [30].

Metronidazole concentrations in the CSF compartment (C_CSF_) were modelled as:(6)$$V_{\mathrm{CSF}}\frac{{\mathrm{dC}}_{\mathrm{CSF}}}{\mathrm{dt}}=C_{\mathrm{brain}\;\mathrm{vasc}}{\mathrm{PS}}_{\mathrm{CSF}}+C_{\mathrm{ECF}}Q_{\mathrm{bulk}}-C_{\mathrm{CSF}}\left({\mathrm{PS}}_{\mathrm{CSF}}+Q_{\sin k}+Q_{\mathrm{EVD}}\right)$$
where Q_EVD_ is the EVD flow determined experimentally during a certain time interval (Appendix A) and used as a covariate to describe the elimination of metronidazole from the CSF compartment. It was assumed that EVD was inserted to reduce intracranial pressure elevation due to the interruption of CSF flow and blockage in CSF reabsorption sites in patients with intracranial hemorrhage [14], such as:(7)$$Q_{\sin k}=Q_{\sin k,\;\mathrm{physio}}-Q_{\mathrm{EVD}}$$

Q_sink,physio_ is the physiological value of Q_sink_ (Table 4). 

Metronidazole concentrations in the EVD compartment (C_EVD_) were modelled for each collection interval as:(8)$$V_{\mathrm{EVD}}\frac{{\mathrm{dC}}_{\mathrm{EVD}}}{\mathrm{dt}}=C_{\mathrm{CSF}}Q_{\mathrm{EVD}}$$
where V_EVD_ is the volume of EVD samples determined experimentally during a certain time interval (Appendix A).

### 4.3. Sensitivity Analysis

A sensitivity analysis was performed to evaluate the impact of various model parameters on CNS concentrations of metronidazole after a single intravenous-infusion of 500 mg over 30 min.

#### 4.3.1. Impact of Brain Pathophysiological Changes

To investigate the impact of pathological changes caused by TBI or SAH on CNS PK profiles of metronidazole, simulations were performed by varying the values of PS_ECF/CSF_, V_ECF_ and Q_brain_ within ranges of 100–500%, 100–140% and 20–100% of their original values, respectively. The values of the selected model parameters were varied one by one while keeping all other parameters fixed to the original values (Table 4), and the resulting plasma, ECF and CSF concentration profiles were plotted.

#### 4.3.2. Impact of EVD

The impact of the EVD on CSF concentrations was investigated by varying Q_EVD_ from 0.001 L/h to 0.04 L/h, these two values corresponding to the minimum and the maximum flows observed during a certain time interval in a particular patient (Appendix A). A simulation without EVD (Q_EVD_ = 0 L/h) was also carried out.

### 4.4. Evaluation of Drug Permeability across the BBB and BCSFB

Metronidazole permeability across the BBB (PS_ECF_) was fixed in the PBPK model to the value predicted by Simcyp from system- and drug-specific parameters. On the other hand, PS_ECF_ was estimated from an in vitro Caco-2 model for comparison purposes. The final protocol is detailed in the Appendix B. Briefly, the apparent permeability of metronidazole was measured in the apical-to-basolateral (P_app,A-B_) and basolateral-to-apical (P_app,B-A_) directions. An efflux ratio was calculated as the ratio of P_app,B-A_ to P_app,A-B_. For a ratio lower than 2, indicating passive diffusion across biological membranes [33], a unique value of P_app_ was calculated as the mean of P_app,A-B_ and P_app,B-A_ and was scaled to PS_ECF,in vitro_ by multiplying by the physiological SA_BBB_ and the brain weight (B_W_) (Table 4):(9)$${\mathrm{PS}}_{\mathrm{ECF},\mathrm{in}\;\mathrm{vitro}}=P_{\mathrm{app}}\times {\mathrm{SA}}_{\mathrm{BBB}}\times B_{W}$$

Permeability across the BCSFB (PS_CSF_) was assumed to be half of PS_ECF_. To validate this hypothesis, PS_ECF_ and PS_CSF_ were estimated by the model.

## 5. Conclusions

Metronidazole was selected as a representative antibiotic with relatively rapid and extensive CNS penetration due to its sufficiently high membrane permeability and absence of affinity for efflux transport systems present at the BBB or BCSFB level to develop a PBPK model from plasma and brain ECF as well as CSF concentrations previously measured in neurocare ICU patients. This model shows that with this type of antibiotic, physio-pathological alterations of membrane permeability PS_ECF/CSF_, cerebral blood flow or brain ECF volume would have no effect on ECF or CSF PK profiles. This work will serve as a starting point for the development of a new PBPK model to describe the CNS distribution of cefotaxime as a representative of antibiotics with more limited CNS distribution due to lower permeability and affinity for efflux pumps, which are frequently used for the treatment of CNS infections. 

## Figures and Tables

**Figure 1 antibiotics-11-01293-f001:**
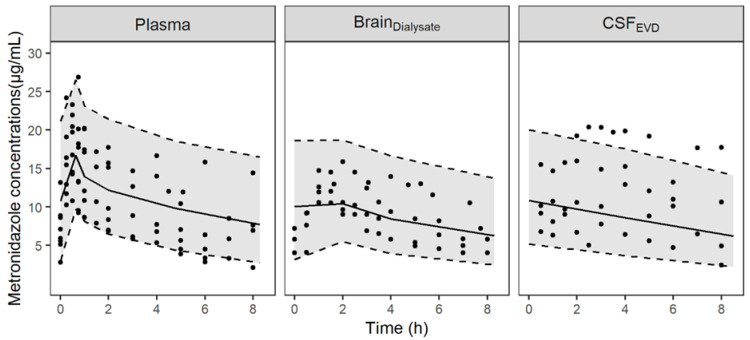
Visual predictive checks of the final PBPK model for unbound metronidazole concentrations in plasma, brain dialysates after correction by in vivo probe recovery (Brain_Dialysate_) and collection bag of the EVD (CSF_EVD_). Circles represent observed data; the solid lines correspond to the median and the grey-shaded area depict the 90% prediction interval delimited by the 5th and 95th percentiles (dashed lines) for 1000 simulated profiles.

**Figure 2 antibiotics-11-01293-f002:**
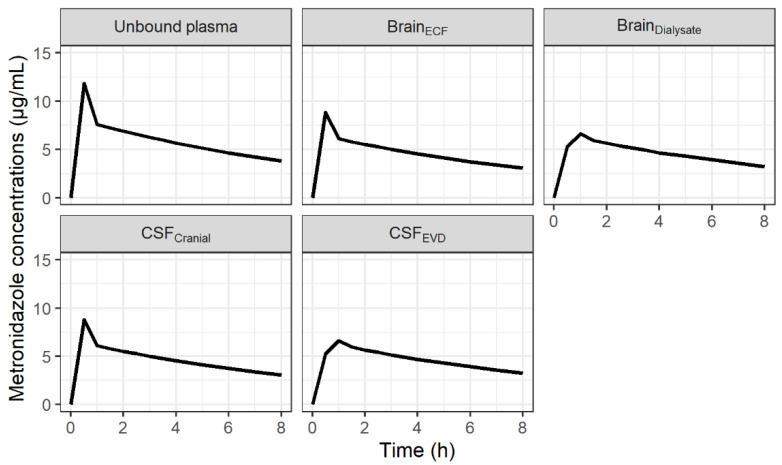
Predicted concentration-time profiles of metronidazole in unbound plasma, brain ECF, brain dialysates after correction by in vivo probe recovery, CSF in the lateral ventricle (CSF_LV_) and CSF collected by extra-ventricular drainage (CSF_EVD_) for a typical patient after administration of 500 mg q8h.

**Figure 3 antibiotics-11-01293-f003:**
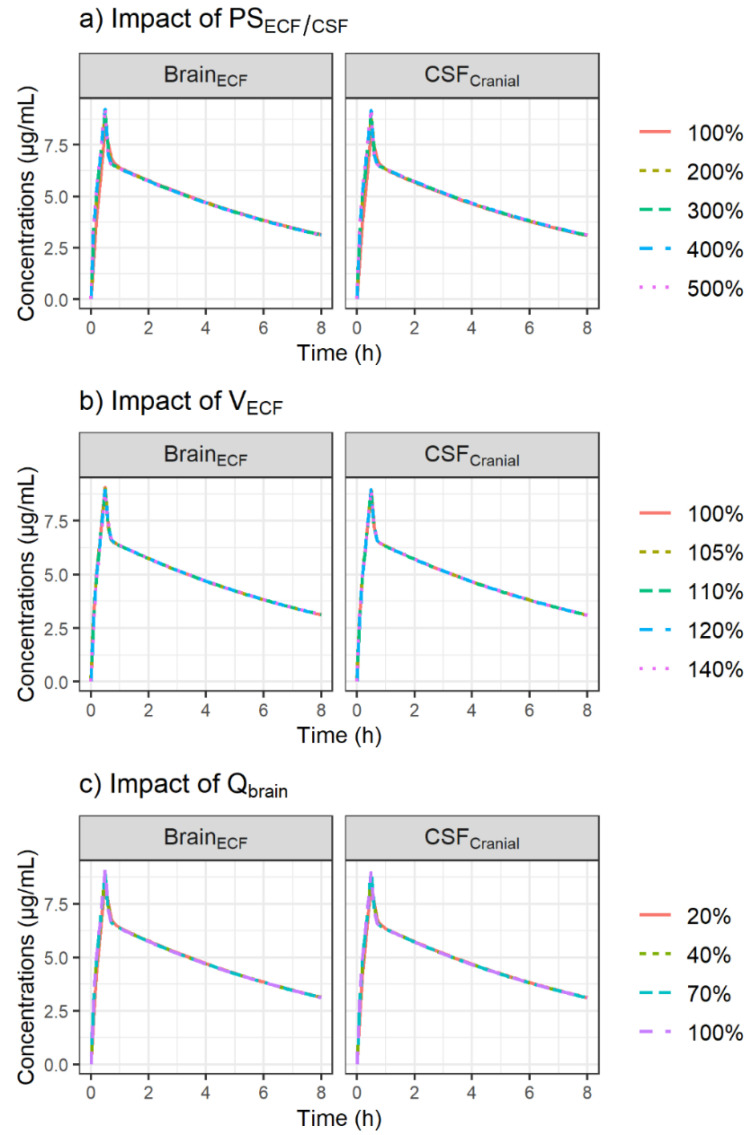
Simulation of the impact of increased passive diffusion clearances, PS_ECF/CSF_ (**a**), increased brain ECF volume, V_ECF_ (**b**) and reduced cerebral blood flow, Q_brain_ (**c**) on CNS PK profiles of metronidazole after administration of a single dose of 500 mg using the minimal PBPK model. The plots were stratified by CNS compartment (panels).

**Figure 4 antibiotics-11-01293-f004:**
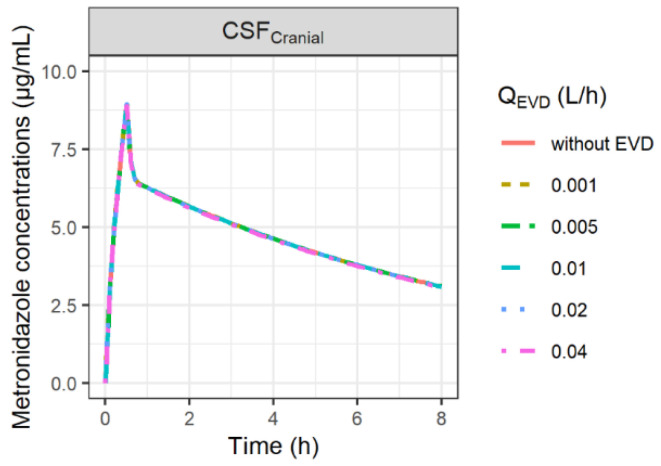
Simulation of the impact of external ventricular drain flow (Q_EVD_) on CSF concentrations of metronidazole. Q_EVD_ was assumed to be constant over time and to restore the physiological CSF outflow (Q_sink_). Thus, when Q_EVD_ was higher than the physiological value of Q_sink_ (Q_EVD_ > 0.024 L/h), Q_sink_ was set at 0 L/h.

**Figure 5 antibiotics-11-01293-f005:**
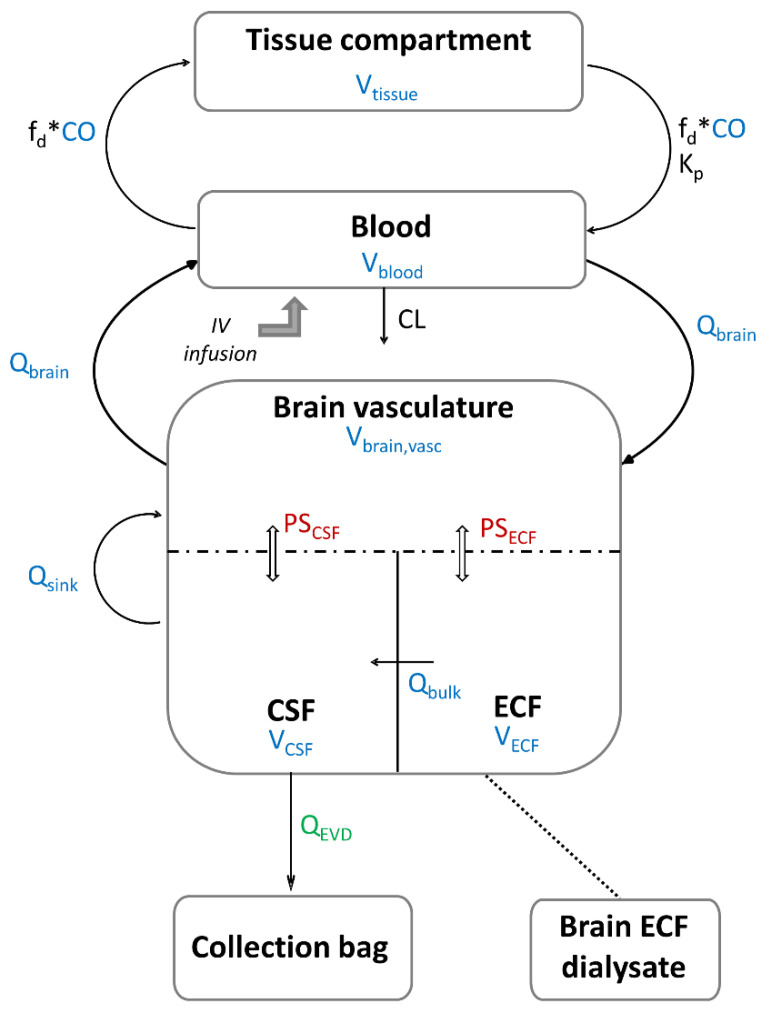
Schematic representation of metronidazole minimal PBPK model: the values of the parameters in color are fixed and those of the parameters in black are estimated. Parameters in blue are system-specific parameters, the parameter in green is determined experimentally, and parameters in red are parameters derived from system- and drug-specific parameters. The dashed line represents the blood–brain barrier (BBB) and the blood–cerebrospinal fluid barrier (BCSFB). ECF: extracellular fluid, CSF: cerebrospinal fluid, CO: cardiac output, Q_brain_: cerebral blood flow, Q_bulk_: bulk flow, Q_sink_: sink flow, Q_EVD_: flow of the external ventricular drain, PS_ECF_: rate of bidirectional passive unbound drug transfer across the BBB, PS_CSF_: rate of bidirectional passive unbound drug transfer across the BCSFB, V_blood_: blood volume, V_tissue_: volume of non-CNS tissue compartment, V_brain,vasc_: brain vascular volume, V_ECF_: ECF brain tissue volume, V_CSF_: cranial CSF volume, fd: fraction of CO going to the non-CNS tissue compartment, K_p_: unbound tissue to blood partition coefficient, CL: total blood clearance.

**Table 1 antibiotics-11-01293-t001:** Parameter estimates for metronidazole in plasma, ECF and CSF.

Parameter	Symbol	Value (95% CI) ^a^	IIV %CV (95% CI) ^a^
Unbound blood partition coefficient for tissue compartment	K_p_	0.796 (0.693–0.923)	-
Fraction of cardiac output going to the non-CNS tissue compartment	fd	0.86 FIX ^b^	
Total blood clearance of unbound drug	CL (L/h)	7.28 (5.77–9.53)	35.2 (24.0–57.8)
Rate of bidirectional passive unbound drug transfer across the BBB	PS_ECF_ (L/h)	6.4 FIX ^c^	
Rate of bidirectional passive unbound drug transfer across the BCSFB	PS_CSF_ (L/h)	3.2 FIX ^d^	
Proportional residual variability for plasma	σ_prop,plasma_ (%)	14.4 (9.74–19.1)	-
Additive residual variability for plasma	σ_add,plasma_ (µg/mL)	1.18 (0.320–2.90)	-
Proportional residual variability for ECF	σ_prop,ECF_ (%)	22.8 (17.5–29.1)	-
Proportional residual variability for CSF	σ_prop,CSF_ (%)	28.2 (22.0–36.5)	-

IIV: inter-individual variability; CV: coefficient of variation; CI: confidence interval; ^a^ The 95% CI was obtained by sampling importance resampling. ^b^ Parameter fixed to the maximum allowed value due to identifiability issue; ^c^ parameter fixed to the value predicted by Simcyp based on system (surface area of BBB) and drug-specific parameters (log P and molecular weight); ^d^ PS_CSF_ was assumed to be half of PS_ECF_.

**Table 2 antibiotics-11-01293-t002:** Comparison of PS values calculated from system- and drug-specific paramaters (PS_calculated_) using Simcyp, estimated by the model (PS_estimated_) or scaled from in vitro Caco-2 permeability parameters (PS_in vitro_).

	PS_calculated_	PS_in vitro_	PS_estimated_
**PS_ECF_ (L/h)**	6.4	8.0	0.904
**PS_CSF_ (L/h)**	3.2 *	4.0 *	0.398

* PS_CSF_ was assumed to be half PS_ECF._

**Table 3 antibiotics-11-01293-t003:** Demographic characteristics of patients.

	Patients							
Parameter	1	2	3	4	5	6	7	8
Age (yrs)	55	64	52	65	73	51	52	34
Ht (cm)	180	172	175	172	178	176	180	175
Sex ^a^	M	M	M	M	M	M	M	M
Wt (kg)	90	90	77	79	90	80	115	75
Creatinine clearance (mL/min) ^b^	171	118	86	144	165	84	141	306
Serum albumin (g/L)	22	NA	31	42	NA	NA	NA	NA
Serum total proteins (g/L)	66	61	64	67	61	53	66	57
Admission type ^c^	TBI	TBI	SAH	TBI	SAH	SAH	VH	SAH
Sample types ^d^	Blood + ECF	Blood + ECF	Blood + ECF	Blood + ECF	Blood + CSF	Blood + CSF	Blood + CSF	Blood + CSF

^a^ M, male; ^b^ calculated by using the MDRD (modification in diet of renal disease) equation; ^c^ TBI, trauma brain injury; SAH, subarachnoid hemorrhage; VH, ventricular hemorrhage; ^d^ ECF, extracellular fluid; CSF, cerebrospinal fluid; NA, not available.

**Table 4 antibiotics-11-01293-t004:** Physiological model input parameters.

Parameter	Definition	Value	Reference
System-Specific Parameters			
Q_bulk_	Bulk flow: flow rate from ECF to CSF	0.0105 L/h	[48]
Q_sink, physio_	Sink flow	0.024 L/h	[48]
CO	Cardiac output	312 L/h	[49]
Q_brain_	Brain blood flow	42 L/h	[2]
V_ECF_	ECF brain tissue volume	0.24 L	[48]
V_CSF_	Cranial CSF volume	0.130 L	[27]
V_blood_	Blood volume	5.85 L	[49]
V_brain,vasc_	Brain vascular volume	0.0637 L	[2]
SA_BBB_	Surface area of the BBB	157 cm²/g brain	[2]
B_W_	Brain weight	1274 g	[2]
**Drug-Specific Parameters**			
MW	Molecular weight	171 g/mol	[50]
log P	Octanol:water partition coefficient	−0.459	[50]
BP	Blood-to-plasma concentration ratio	0.82	PK-Sim prediction

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
