# Peer review of "A Minimal Physiologically Based Pharmacokinetic Model to Characterize CNS Distribution of Metronidazole in Neuro Care ICU Patients"

_antibiotics, 2022, doi:10.3390/antibiotics11101293_

Round 1

Reviewer 1 Report

Overall, the paper presents a significant contribution to knowledge. Caco-2 cells has been demonstrated to be useful for this experimental design, given that PK experimentation is often challenging especially when it involves absorption and distribution.

While the rest of the paper seems acceptable in their form in my opinion, my only recommendation is to also add more information on the quantification method of Metrodinazole. Specifically, this section paper would be highly improved if the following are also mentioned or shown (you may add in the paragraph in lines 563-568):

1. Limit of Detection / Limit of Quantification of the HPLC-UV method used

2. Linear range (Please show a calibration curve in your supplemental data, and R^2 value)

3. At least 1 chromatogram of a standard and an unknown and internal standard (important to demonstrate specificity of your method to Metrodinazole)

Below are recommended but not required, but would help future researchers should your paper be referenced or compared/cited for applications of HPLC-UV methods to quantify Metrodinazole:

1. Recovery data of the method (i.e. how much of a known Metrodinazole was recovered from your extraction)

2. Any quality assurance measures you used (i.e. quality control accuracy and precision, to demonstrate that your HPLC-UV assay works within reasonable accuracy/precision)

This information will help your paper have a more solid justification in terms of the validity of the method you used to quantity Metrodinazole.

Reviewer 2 Report

 Give fold error in simulated vs observed PK parameters.

Please acknowledge that the work has been previously shared as a poster. https://hal.archives-ouvertes.fr/hal-02528416

Figure 3/4: Discuss why there were no differences in concentration for sensitivity analysis.

 How was the extent of neuro-damage tabulated for the study?
